# Look at What I'm Doing: Self-Supervised Spatial Grounding of Narrations in Instructional Videos

**Reuben Tan**[1]    **Bryan A. Plummer**[1]    **Kate Saenko**[1,2]    **Hailin Jin**[3]    **Bryan Russell**[3]

[1]Boston University, [2]MIT-IBM Watson AI Lab, IBM Research, [3] Adobe Research

{rxtan, bplum, saenko}@bu.edu, {hljin, brussell}@adobe.com

https://cs-people.bu.edu/rxtan/projects/grounding_narrations

## Abstract

We introduce the task of spatially localizing narrated interactions in videos. Key to our approach is the ability to learn to spatially localize interactions with self-supervision on a large corpus of videos with accompanying transcribed narrations. To achieve this goal, we propose a multilayer cross-modal attention network that enables effective optimization of a contrastive loss during training. We introduce a divided strategy that alternates between computing inter- and intra-modal attention across the visual and natural language modalities, which allows effective training via directly contrasting the two modalities' representations. We demonstrate the effectiveness of our approach by self-training on the HowTo100M instructional video dataset and evaluating on a newly collected dataset of localized described interactions in the YouCook2 dataset. We show that our approach outperforms alternative baselines, including shallow co-attention and full cross-modal attention. We also apply our approach to grounding phrases in images with weak supervision on Flickr30K and show that stacking multiple attention layers is effective and, when combined with a word-to-region loss, achieves state of the art on recall-at-one and pointing hand accuracies.

## 1   Introduction

Content creators often add a voice-over narration to their videos to point out important moments and guide the viewer on where to look. While the timing of the narration in the audio track in the video gives the viewer a rough idea of when the described moment occurs, there is not explicit information on where to look. We, as viewers, naturally infer this information from the narration and often direct our attention to the spatial location of the described moment.

Inspired by this capability, we seek to have a recognition system learn to spatially localize narrated moments without strong supervision. We see a large opportunity for self-supervision as there is an abundance of online narrated video content. In this work, we primarily focus on spatially localizing transcribed narrated interactions in a video, illustrated in Figure 1 (right). Unlike prior phrase grounding work (Figure 1 left), which primarily focuses on matching a noun phrase to an object, our task involves matching entire sentences to regions containing multiple objects and actions.

Not only is this task integral to advancing machine perception of our world where information often comes in different modalities, it also has important implications in fundamental vision-and-language research such as robotics, visual question answering, video captioning, and retrieval. Our task is challenging as we do not know the correspondence between spatial regions in a video and words in the transcribed narration. Moreover, there is a loose temporal alignment of the narration with the described moment and not all words refer to spatial regions (and vice versa). Finally, narrated interactions may be complex with long-range dependencies and multiple actions. In Figure 1 (right), notice how "them" refers to the onions and that there are two actions – "cut" and "add".

35th Conference on Neural Information Processing Systems (NeurIPS 2021).

"**A dog** playing with a **blue ball**."

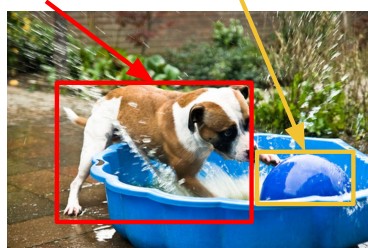

"**Cut the onions and add them to the tray.**"

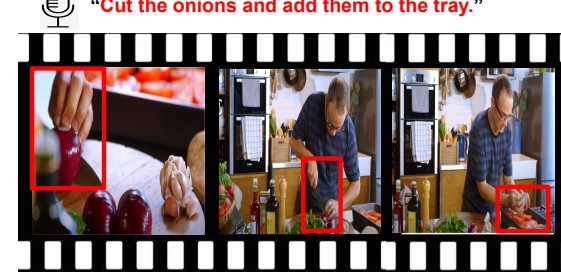

Phrase Grounding (prior work)                    Narrated Interaction Localization (this paper)

Figure 1: Prior work on *phrase grounding* focuses on localizing single objects in static images (left, example from Flickr30K [40]). In this paper, we propose a new task of spatially *localizing narrated interactions* in video (right). In the task of phrase grounding, the goal is to localize nouns or adjective-noun phrases in images. In contrast, interaction grounding may involve multiple objects and actions and use entire sentences. Our approach self-trains on a large corpus of narrated videos and does not require supervision. (Video credit: Woolworths [57])

Contrastive learning [38] and computing contextual features via an attention model [52] are natural tools for addressing the aforementioned alignment and long-range cross-modal dependency challenges. While contrastive learning has been successfully applied for temporal alignment of a video clip with a narration [35], the alignment of spatial regions with narrations has not been addressed. Naively applying a joint attention model over the set of features for the two modalities, followed by optimizing a contrastive loss over the two modalities' aggregated contextual features, results in a fundamental difficulty. Recall that an attention computation comprises two main steps: (i) selecting a set of features given a query feature, and (ii) aggregating the selected features via weighted averaging. By selecting and aggregating features over the two modalities during the attention computation and subsequently optimizing a contrastive loss, the model in theory may select and aggregate features from only one modality to trivially align the contextual representations.

While there are approaches for grounding natural language phrases in images without spatial supervision, the best-performing approaches involve matching individual words to regions [22] or not using any paired text-image supervision at all [54]. This strategy is effective for simpler short phrases, as illustrated in Figure 1 (left). However, consider the interaction "cut the onions and add them to the tray", shown in Figure 1 (right). This interaction is described with more complex, compositional language. In the first two frames, there are multiple "onion" objects visible and the correct one depends on the action being applied. Unlike the task of object grounding, it is not sufficient to simply co-localize all of the mentioned objects. We will show that it is difficult to effectively employ a strategy that matches individual words to regions to localize such complex interactions.

To address these challenges, we make the following contributions. First, we learn to localize the spatial location of a narrated interaction from abundantly available narrated instructional videos [36]. Our approach does not require manually collecting sentence descriptions or the locations of the described interactions for training. Here, the automatically transcribed narrations allow for self-supervised learning via alignment with the video clip.

Second, we propose a new approach for directly contrasting aggregated representations from the two modalities while computing joint attention over spatial regions in a video and words in a narration. We show that optimizing a loss over the aggregated, sentence-level representation allows for a global alignment of the described interaction with the video clip, and offers an improvement over optimizing a matching loss over words and regions. To overcome the network learning a trivial solution while directly contrasting jointly attended features from the two modalities during training, we design network attention layers that do not allow feature aggregation across the two modalities. Our strategy involves alternating network layers that compute inter- and intra-modal attention. Our inter-modal attention layer allows features from one modality to select and aggregate features from only the other modality, and not within its own modality. Our intra-modal attention layer selects and aggregates features from within the same modality. This strategy ensures that the output representations do not aggregate features across the two modalities. In combination with stacking the inter- and intra-modal attention layers, our approach attends jointly and deeply and directly contrasts the resulting contextual representations from the two modalities.

Finally, we introduce an evaluation dataset that provides bounding box annotations for interactions described by natural language sentences. Our dataset is built on the validation split of the YouCook2 dataset [65] and contains approximately 1000 segments of varying durations. We demonstrate our approach on our collected evaluation dataset of localized interactions and on localizing objects via the YouCook2-BB benchmark where we show that we outperform shallow and full cross-modal attention. We also apply our approach to grounding phrases in images with weak supervision on Flickr30K and show that stacking multiple attention layers is effective with our loss. Furthermore, we show that our loss is complementary with the word-to-region loss of Gupta *et al*. [22], and when combined with it, achieves state of the art on recall-at-one and pointing-hand accuracies.

## 2   Related Work

**Self-supervised learning.**   There has been significant progress in natural language processing, resulting in effective and robust learned word representations [37, 17, 60, 39] that achieve state-of-the-art performance on downstream tasks. Recently, it has also garnered a lot of interest in the computer vision community, achieving state-of-the-art performance in unsupervised pre-training of deep visual models. [10, 24, 11] have demonstrated that effective image representations can be learnt simply by contrasting between augmented views of the same image without labels. [62, 45] found that the resulting representations are biased by background pixels and tried to improve foreground object localization via data augmentation or saliency. In video, self-supervised tasks like pace prediction ad future prediction [53, 23, 19, 26, 6, 55] were used for pre-training, while [2] used self-supervision on audio-visual data for tasks like unsupervised speaker localization. In this work, we propose a self-supervised method to spatially localize activity descriptions in video.

**Object and action localization.**  Approaches generally leverage a region proposal network (RPN) [51, 21, 20] as well as region-based convolutional neural network (CNNs) to detect objects or localize actions temporally [59] or spatio-temporally [27]. However, they are often trained on curated datasets and are, consequently, limited by a fixed predefined number of object categories in these datasets. [65] collected a dataset of cooking videos labeled with descriptions and temporal segments but did not address spatial grounding of activities. Motivated by this limitation, our proposed approach aims to learn to recognize human interactions with objects that belong to the long-tailed distribution from uncurated and unlabeled online videos. There have been video datasets collected with bounding box annotations with associated natural language descriptions [63]. These datasets provide bounding boxes for noun phrases [15, 63] or for egocentric videos [15]. We collect the first dataset of captioned interactions annotated with spatial bounding boxes for evaluating narrated interactions.

**Vision-and-language.**  Semantic information from natural language has often been exploited to provide an additional source of supervision for learning visual representations. Such approaches have generally leveraged image [29, 31, 40] and video [5] datasets that are annotated with natural language descriptions to learn a joint embedding space, where visual and language representations of semantically-similar pairs are close. Prior approaches have mainly focused on retrieving short clips from a large video corpus [35] or localizing relevant segments within untrimmed videos [5, 9, 12]. However, they do not identify the relevant spatial locations of the described interactions. Some existing works aim to spatially localize a natural language query [44, 61, 30] but focus on objects and train on ground truth bounding boxes, whereas our approach is unsupervised. Object grounding with weak or no supervision is addressed in [12, 4, 7]. To alleviate the costly annotations of curated datasets, [35] have proposed to learn robust video representations as well as an effective joint video-text embedding space from a large corpus of unlabeled and uncurated instructional videos [36]. In concurrent work, CLIP [41] trained image-text representations on a dataset of 400 million (image, text) pairs collected from the internet. These efforts to learn from "free" data are similar in spirit to our work, however, we focus on the localization task.

## 3   Self-Supervised Grounding of Narrated Video Interactions

Given a video clip and a narration, our objective is to learn to localize the relevant spatial regions in the clip that correspond to the interaction described in the narration without localization supervision, *e.g.*, bounding boxes or segmentation masks. To this end, we propose a novel Contrastive Multilayer Multimodal Attention (CoMMA) module that allows for attention interactions between the features

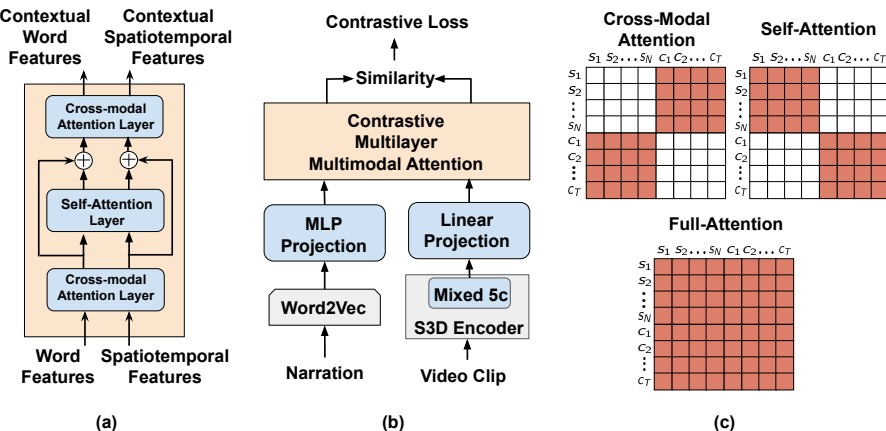

Figure 2: **Approach overview.** (a) Our proposed **Contrastive Multilayer Multimodal Attention (CoMMA)** module. (b) Our proposed architecture for computing joint attention over narrations and videos. (c) Binary masks for computing the different types of cross-modal attention. The shaded regions represent "true" Boolean values where a query feature may select a key. If the mask value is "false", then that corresponding (query, key) pair is not considered in the computation of attention and weighted feature aggregation.

for spatiotemporal regions and words, as well as an effective contrasting of the two modalities' signals. We define two modalities to have attention interactions when their features attend to each other to compute their final contextualized representations. This setup is in contrast to when the two modalities' features only interact when the training loss is computed [35].

Our CoMMA module comprises alternating bidirectional cross-modal attention and self-attention layers (Figure 2(a)). The intuition underlying our proposed approach is that a hierarchical multimodal attention model with multiple stacked layers will encourage a fine-grained alignment between video spatiotemporal regions and narration words and ignore irrelevant regions/words. More specifically, each bidirectional cross-modal attention layer computes new representations for a target modality with the latent representations from the source modality to learn the relevance of each spatiotemporal region to each word or phrase, and vice versa. Additionally, the self-attention layer serves to identify relevant words and regions by aggregating contextual information between the augmented unimodal features. Through repeated attention interactions of the two modalities and aggregation of unimodal contextual information, our CoMMA module becomes more discriminative of irrelevant regions and words. Moreover, our module allows for applying a contrastive loss to the final representations for the two modalities since there is no cross-modal feature aggregation, *i.e.*, a modality may self-select features or select features from another modality by attending to them, but it may never select features from both modalities. Our approach goes beyond shallow attention architectures [3] and applying a contrastive loss to independent fixed-length features for each modality (*i.e.*, no attention interaction between the two modalities) [35]. Our multimodal attention module can be easily stacked on top of base feature encoders for our task (Figure 2(b) shows our full network). Next, we describe our CoMMA module in detail, how we employ a contrastive loss for training, and how we employ our network for the localization task.

### 3.1 Contrastive Multilayer Multimodal Attention

We formulate each attention layer in our CoMMA module as a key/query/value attention mechanism that is commonly used in attention models [52]. Our module accepts a concatenated sequence of narration word and video spatiotemporal region features and a set of binary attention masks. Let $c_r$ be a spatiotemporal feature for region $r$ and $s_j$ be a word feature for the $j$-th word in a narration with $N$ words. We obtain features $c_r$ and $s_j$ via base feature encoders that are projected to a common embedding space using linear and MLP layers. We form matrix inputs $C_0$ and $S_0$ by stacking the spatiotemporal and word features $c_r$ and $s_j$, respectively, as column vectors.

In Figure 2(c), the shaded regions of the binary masks indicate which features interact during the attention computation for a layer. Specifically, the cross-modal attention (CA) mask $M^{CA}$ computes the attention over the words with respect to each region and vice versa. Note that this setup differs

from the full-attention (FA) mask $M^{FA}$ (illustrated in Figure 2(c) bottom) that computes attention over the *entire* multimodal sequence of words and regions and the self-attention (SA) mask $M^{SA}$ (illustrated in Figure 2(c) top-right) that computes attention within each modality. We extend the mechanism of Vaswani *et al.* [52] by incorporating a mask that modulates the attention computation. Formally, let $K$, $Q$, $V$ be the matrices of features for the keys, queries and values, respectively (the individual features are stacked as column vectors). Let the masked attention mechanism be represented as:

$$Attn(K, Q, V, M) = V \operatorname{softmax}\left(\frac{(Q^\top K) \odot M}{\sqrt{D}}\right) \qquad (1)$$

where $M$ is any one of the binary masks shown in Figure 2(c), $D$ denotes the scalar dimensionality of the query features, and $\odot$ is the Hadamard product.

For simplicity, let matrix $Y_0^{SA} = [C_0, S_0]$ denote the stacked input features from both modalities. In addition, let $W_{K,l}^i$, $W_{Q,l}^i$ and $W_{V,l}^i$ denote the projection matrices for keys, queries, and values, respectively, for attention type $i$ and layer $l$. Here, $i$ can be used to represent a cross-attention (CA) or self-attention (SA) layer and $l$ is the layer number. We compute the output of the cross-modal attention layer by passing in the initial input features or contextualized feature outputs of the self-attention layer:

$$Y_{l+1}^{CA} = CrossAttn(Y_l^{SA}) = Attn(W_{K,l+1}^{CA} Y_l^{SA}, W_{Q,l+1}^{CA} Y_l^{SA}, W_{V,l+1}^{CA} Y_l^{SA}, M^{CA}) \qquad (2)$$

where $M^{CA}$ is the cross-modal attention mask. Note that if the input to Equation (2) is $Y_l^{SA} = [C_l, S_l]$, then the output will be $Y_{l+1}^{CA} = [S_{l+1}, C_{l+1}]$, *i.e.*, (contextualized) video features become (contextualized) language features and vice versa. We emphasize that in the cross-attention layer, there is no mixing of features from different modalities where an elementwise sum is applied to features across the different modalities. Consequently, it allows both modalities to attend to each other without leaking any information between them. We include a mathematical formalization of this operation in the supplementary. The similarity scores computed between the queries and the keys as the scaled dot product in Equation (1), act as a soft attention mechanism to measure the relevance of each word with respect to a region. The softmax-normalized scores are multiplied with the value (words) vectors to compute a region-specific narration representation. To illustrate this concept, consider a YouTube video clip about cooking fried chicken. The words 'fried' and 'cook' will have a higher similarity score to regions in the video clip than 'subscribe' since some regions will be highly relevant to the former. Conversely, the modalities of the keys and queries can be swapped to compute word-specific clip representations.

To aggregate contextual information over the augmented representations of the two modalities, the outputs of the cross-modal attention layer are passed into a self-attention layer. The self-attention mask $M^{SA}$ ensures that attention is only computed between elements of the same modality. Formally, this computation can be represented as

$$Y_l^{SA} = SelfAttn(Y_l^{CA}) = Attn(W_{K,l}^{SA} Y_l^{CA}, W_{Q,l}^{SA} Y_l^{CA}, W_{V,l}^{SA} Y_l^{CA}, M^{SA}) + Y_l^{CA} \qquad (3)$$

where $M^{SA}$ is the self-attention mask. Its output is passed into the next cross-attention layer. We denote the final output of our module after $L$ layers as $C_L$ and $S_L$ where $C_L \in \mathbb{R}^{D \times T}$ is the set of contextualized spatiotemporal region features and $S_L \in \mathbb{R}^{D \times N}$ is the set of contextualized word representations.

**Comparison to multimodal attention models.** To evaluate the importance of multimodal attention modules for localizing narrations, we briefly compare CoMMA against other state-of-the-art variants. Akbari et al. [3] is an image-level approach that utilizes features from different levels of an encoder. However, unlike CoMMA, it only utilizes a single round of cross-modal interaction. Another highly relevant work is the Contrastive Bidirectional Transformer (CBT) [50]. Despite the similarities in attention layers, the CBT model computes self-attention over the entire multimodal sequence (similar to the full-attention mask in Figure 2(c)) and aggregates a summary knowledge into a sentinel vector. In contrast, CoMMA allows for repeated attention interactions without mixing of features from different modalities. We show in Section 4 that this mechanism is critical for localization.

### 3.2  Contrastive loss and inference

We train our proposed multimodal attention module by contrasting between positive and negative pairs of video clips and narrations. Formally, we aim to learn language-aligned visual representations

for video clips such that features for corresponding videos and narrations are similar and non-corresponding features are dissimilar. $C_{L,r}$ denotes the column vector for the final contextualized feature for region $r$ and $S_{L,j}$ denotes the column vector for the final contextualized representation for word $j$. The final video clip representation $\hat{C}$ and narration sentence representation $\hat{S}$ are computed by mean-pooling over the spatiotemporal regions $\hat{C} = \frac{1}{R}\sum_{r=1}^{R} C_{L,r}$ and words $\hat{S} = \frac{1}{N}\sum_{j=1}^{N} S_{L,j}$ respectively. Let $\left(\hat{C}^{(i)}, \hat{S}^{(i)}\right)$ be the $i$-th training example pair. We adapt the InfoNCE loss [38] by defining the sentence loss $\mathcal{L}_{sent}$ as the sum of log ratios over self-training pairs:

$$\mathcal{L}_{sent} = -\sum_{i=1}^{n} \log \left( \frac{\exp\left(\hat{C}^{(i)} \cdot \hat{S}^{(i)}\right)}{\exp\left(\hat{C}^{(i)} \cdot \hat{S}^{(i)}\right) + \sum_{m \sim \mathcal{N}_{i,S}} \exp\left(\hat{C}^{(i)} \cdot \hat{S}^{(m)}\right) + \sum_{m \sim \mathcal{N}_{i,C}} \exp\left(\hat{C}^{(m)} \cdot \hat{S}^{(i)}\right)} \right) \tag{4}$$

where the negative sets $\mathcal{N}_{i,C}$ and $\mathcal{N}_{i,S}$ comprise indices for non-corresponding video clip and narration pairs for the $i$-th training sample, and $n$ denotes the total number of training samples. Note that our CoMMA module is applied to negative pairs as well. In contrast to the word-level objective (defined in Section 4) used in [22] which contrasts word features separately, we empirically find that contrasting aggregated features in the sentence loss $\mathcal{L}_{sent}$ better handles when the temporal alignment between narrations and videos is noisy.

**Inference.** During inference, given a natural language sentence and a video clip, we aim to localize the salient spatial regions across the frames in the clip. Beginning from the attention weights of the last cross-modal attention layer, we apply attention rollout [1] to obtain the final attention heatmap over all spatiotemporal regions. In attention rollout, the attention weight matrices from all cross-attention and self-attention layers are multiplied recursively to yield the output localization scores $A$ for each spatiotemporal region, where $A = \prod_{l=0}^{L} W_l$ for attention weights $W_l$ from the $l$-th layer. The resulting localization scores aggregate the total amount of attention by the entire set of spatiotemporal and word features assigned to a query feature.

## 4   Experiments

**Word-level loss.** We experiment with a word-level loss that is used in state-of-the-art phrase grounding models [22, 3], which aims to learn an alignment between each word in the narration and all spatial regions. Gupta *et al.* [22] has demonstrated that the word-level loss is effective at localizing noun phrases in images, which seeks to maximize the mutual information between each word and region features for corresponding video clip and narration pairs. We also consider an extension of the word-level loss where we incorporate the output contextualized representations from our CoMMA module. Let $N$ denote the number of words in the narration, and $n$ the total number of training samples. $S_{L,j}^{(i)}$ denotes the representation output of our CoMMA for the j-th word of the i-th training sample. Finally, for a given input word $S_{0,j}^{(i)}$, we compute its value representation with a multilayer perceptron (MLP): $\bar{S}_j^{(i)} = MLP(S_{0,j}^{(i)})$. Then, the word-level loss is formulated as the sum of log ratios:

$$\mathcal{L}_{word} = -\sum_{i=1}^{n}\sum_{j=1}^{N} \log \left( \frac{\exp\left(S_{L,j}^{(i)} \cdot \bar{S}_j^{(i)}\right)}{\exp\left(S_{L,j}^{(i)} \cdot \bar{S}_j^{(i)}\right) + \sum_{m \sim \mathcal{N}_i} \exp\left(S_{L,j}^{(m)} \cdot \bar{S}_j^{(i)}\right)} \right) \tag{5}$$

where the negative set $\mathcal{N}_i$ of the $i$-th training sample comprises indices for the word features that are attended to by non-corresponding video clips.

### 4.1   Grounding interactions and objects in video

**Self-training and evaluation datasets.** We self-train our proposed model on instructional videos from the HowTo100M dataset [36]. To reduce the computational burden of training over the entire 100M clips, we identify a set of clips from the dataset that roughly aligns with the YouCook2 dataset. To achieve this goal, we extract a list of verb and nouns from the vocabulary of YouCook2 and

filter HowTo100M video clips with narrations that contain these words. Our final pre-training set comprises approximately 250,000 video clips. We use the publicly available base video and language feature encoders [35] that are trained on the entire HowTo100M dataset, which provides a good initialization for our learning task.

For evaluation, we introduce a new evaluation dataset, YouCook2-Interactions, that provides spatial bounding box annotations for interactions that are described by a natural language sentence. Our dataset is built on the validation split of the YouCook2 dataset [65]. We describe our dataset in full in the supplemental. Please see more details of the self-training and our collected dataset in the supplemental. We evaluate using our collected YouCook2-Interactions dataset for localizing narrated interactions.

In addition to localizing interactions given a sentence narration, we also consider localizing objects in video given a single object keyword. For localizing single objects, we evaluate on the YouCook2-BB dataset [64]. The YouCook2-BB dataset augments the original YouCook2 dataset with bounding box annotations for object locations. In our experiments on YouCook2-BB, we only evaluate our approach on the validation split since the test split annotations are not readily available.

**Evaluation criteria.** For both video tasks, we evaluate the quality of the output detections using the pointing hand accuracy criterion. Specifically, given the ground-truth bounding box, we consider our model to have a "hit" if the pixel with the highest co-attention similarity score lies within the box. Otherwise, it is a "miss". The final localization accuracy is computed as the ratio of hits to the total number of hits and misses $\frac{\text{\# hits}}{\text{\# hits} + \text{\# misses}}$.

**Implementation details.** We use publicly available implementations for the separable 3D CNN-gated (S3D-G) encoder [58] and a shallow language encoder [35] built on top of Word2Vec [37] embeddings for our video and language models, respectively. To train our proposed model, we set a learning rate of 1e-4 and optimize the model using the AdamW optimizer [32] with one-epoch linear warmup. In our experiments, we also explore using two CoMMA modules but, due to the large memory requirements, the batch size has to be reduced significantly. Results from our initial experiments and prior work on contrastive learning show that a large batch size is crucial to achieving strong performance. More details are in the supplemental.

During inference, since the temporal and spatial dimensions of the attention heatmap have been downsampled from the original input resolution, the localization scores are temporally and spatially interpolated back to the input resolution. We return the final spatiotemporal location as the mode over all output scores.

**Baselines.** In our evaluations, we compare our approach against center prior, optical flow, and state-of-the-art image-ground baselines. In the center prior baseline, we return the center pixel for each frame. This is a simple yet competitive baseline that performs well due to the nature of these videos, where the main subject is generally in the center of the camera view. As most of the interactions involve motion in cooking videos, we compute an optical flow baseline. For this baseline, we generate an optical flow map for each frame using the OpenCV implementation of the Lucas-Kanade optical flow estimation method [8, 33]. Finally, we select the pixel with the highest magnitude as the location prediction.

Additionally, we compare against four cross-modal attention baselines. First, we compare against the model of Miech *et al.* [35] adapted to our task ("MIL-NCE"). We adapt their publicly available pretrained model by removing the penultimate global average pooling layer in the S3D-G video encoder. Consequently, the output of the video encoder is a set of spatiotemporal features that are projected into the joint embedding space. Then, we compute the dot product between the sentence feature and the spatiotemporal features before performing linear interpolation to determine the mode pixel. Second, we compare against a baseline that computes full attention and leverages a sentinel vector ("Full attention + sentinel") to predict the similarity of the two modalities' features; an MLP is used to convert the output contextualized sentinel vector into a scalar MI-score which is then passed to an InfoNCE-like loss (inspired by CBT [49]). Third, we compare our multilayer model against a shallow cross-modal attention layer ("Shallow attention") inspired by Akbari *et al.* [3]. To maintain fair comparisons, we implement a similar version of their model except we use for the base features the convolutional feature map outputs of the S3D video encoder. Finally, we include comparisons

Table 1: **Interaction localization evaluation on our YouCook2-Interactions dataset.** Our approach outperforms baselines, including shallow and full attention.

| Approach | Training Loss | Localization Accuracy (%) |
|---|---|---|
| Center Prior | - | 31.81 |
| Optical flow | - | 18.75 |
| MIL-NCE [35] | Sentence level | 27.18 |
| Contrastive phrase grounding [22] | Word level | 24.04 |
| Shallow Attention | Sentence level | 48.30 |
| Full attention + sentinel | Sentence level | 39.03 |
| CoMMA (Ours) | Word level | 34.91 |
| CoMMA (Ours) | Sentence level | **55.80** |

Table 2: **Spatiotemporal self-attention ablation and attention layer ablation.** (Left) We report an ablation of the different attention module components in our model. We find that multiple stacked layers outperform shallower architectures. (Right) We ablate different types of self-attention over spatiotemporal features and find computing full self-attention over spatiotemporal features works best.

| Cross-Modal Attention 1 | Cross-Modal Attention 2 | Self-Attention | Localization Accuracy (%) |
|---|---|---|---|
| ✓ | ✗ | ✗ | 48.30 |
| ✓ | ✓ | ✗ | 51.09 |
| ✓ | ✓ | ✓ | **55.80** |

| Self-Attention Type | Localization Accuracy (%) |
|---|---|
| Spatial | 53.88 |
| Temporal | 52.72 |
| Spatial + Temp. | 52.15 |
| Spatiotemporal | **55.80** |

against a state-of-the-art phrase grounding model [22]. We apply the model out of the box to our task with the exception of using our features for fair comparison.

**Interaction localization.** We report in Table 1 results of our approach and the baselines on the task of interaction localization. Our proposed approach outperforms the center prior baseline by a significant margin of 20% on the task of localizing interactions given narration sentences. Surprisingly, computing the region with the maximum degree of motion through optical flow does not provide a good proxy for localization. The large performance gain achieved by our proposed approach over the optical flow baseline suggests that tracking regions with maximum movement alone is insufficient for our task. In addition, we observe that our proposed approach outperforms a state-of-the-art phrase grounding model [22], which is trained via optimizing the word-level loss (Equation 5), by a large margin. One possible reason is that enforcing an alignment between words and regions is not suitable for grounding interactions in videos, especially when there is a weak temporal alignment between clips and narrations. This is corroborated by a huge drop of 20% in localization accuracy when our CoMMA module is trained with the word-level loss instead.

The difference in performance obtained by our model and the MIL-NCE baseline suggests that a model trained for retrieval may not be focusing on the relevant regions that are described by the sentence narration. Our proposed approach outperforms the "full attention + sentinel" baseline by 15%. This result suggests that aggregating contextual information into a sentinel vector causes the model to lose fine-grained information required for localization. We note that, similar to Sun *et al.* [49], this baseline outputs a scalar value that is plugged in directly to the InfoNCE loss; it does not compute a cosine similarity between the video clip and natural language query contextualized features. We hypothesize that computing this MI-like score (instead of directly contrasting the two modalities' features) may somewhat mitigate this baseline from catastrophic failure where features from both modalities are aggregated during the attention interaction process.

**Multilayer attention and spatiotemporal self-attention ablations.** We report in Table 2 (left) an ablation over stacking the different attention layers in our network. Notice that the localization accuracy increases as we stack more attention layers in our network, suggesting that deeper cross-modal attention models help to improve the alignment between the different modalities. We report in Table 2 (right) results from an ablation of the different forms of self-attention over spatiotemporal regions. It is interesting to observe that by applying self-attention to spatial regions alone, the model performs competitively by obtaining a localization accuracy of 53.88%. We obtain our best model

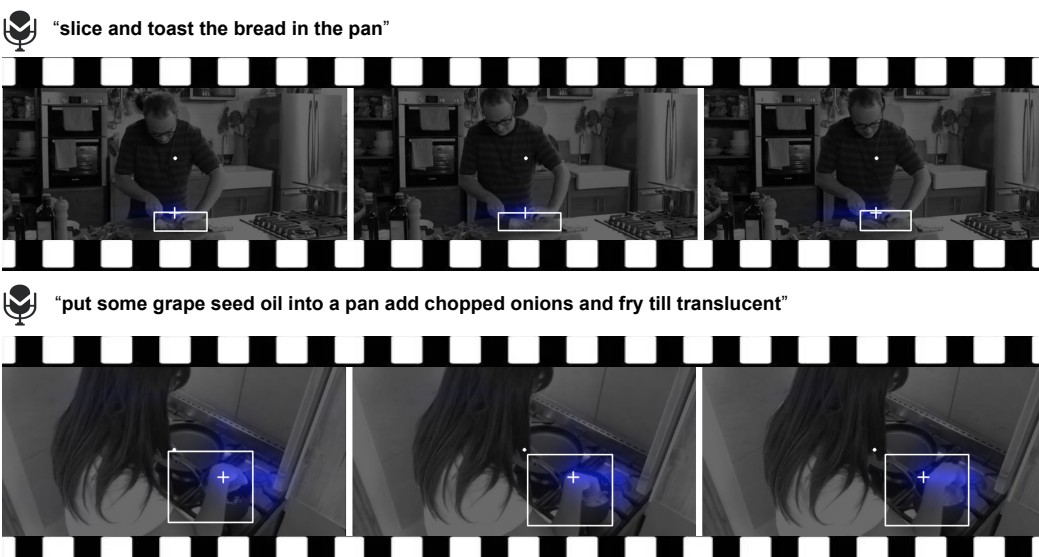

Figure 3: Examples of correct localization predictions by our model. Our approach localizes narrated interactions involving a person's hands and interacted objects. The boxes in these visualizations denote the ground-truth bounding boxes for the transcribed narrations. The white circle and cross indicate the locations of the center prior and the mode pixel of the computed attention score map, respectively. Finally, the blue regions indicate the regions that our CoMMA module determines to be most relevant to the transcribed narrations. (Video credit: Woolworths [57] and Alicia Kirby [28])

when self-attention is computed over all spatiotemporal regions. This finding suggests that reasoning about temporal context is important but it has to be done in conjunction with spatial context reasoning.

**Object localization.** Besides localizing full narrated sentences, we also evaluate our model's capability to localize single objects. Specifically, we compare our approach against the state-of-the-art "not all frames are equal" (NAFAE) model [47]. We report results in Table 3. Despite being trained with a sentence-level loss, we observe in Table 3 that our approach successfully localizes objects corresponding to single noun words. Notably, our proposed approach outperforms the NAFAE baseline. We give more details of the NAFAE baseline in the supplemental.

**Qualitative results.** We show qualitative results in Figure 3. Each frame is overlaid with the computed attention score map to depict the visual cues that our model uses to make its prediction. The white circle indicates the location of the center prior and the cross denotes the mode pixel of the computed attention score map. Notice how our approach correctly localizes the described interaction.

Table 3: **Object localization evaluation on the YouCook2-BB dataset.** By self-training on narrations, our approach learns to localize objects and outperforms the baseline.

| Approach | Full Loc. (%) |
|----------|---------------|
| NAFAE [47] | 46.95 |
| Ours | **59.25** |

### 4.2 Weakly supervised phrase grounding in images

We evaluate our approach on the task of weakly supervised phrase grounding to determine its capability to generalize to localizing noun phrases in images. For this task, we adopt the same setup as in prior work [22, 16, 18] by training on MSCOCO [31] and evaluating on Flickr30k [40].

We report results on the standard recall-at-K (R@K) and pointing hand recall accuracy criteria [22]. For training, we optimize the sum of the word-to-region matching loss and our sentence loss: $\mathcal{L}(\theta) = \mathcal{L}_{word} + \lambda \mathcal{L}_{sent}$. We set $\lambda$ to be 0.005 in our experiments. We compare our approach against state-of-the-art phrase grounding models. Details of how the model in [22] is extended with our CoMMA architecture as well as a description of the baselines are included in the supplemental. We note that the value projection layer for region features in [22] is omitted to incorporate CoMMA. Finally, we build on the same set of visual and language features from [22] for fair comparison.

Table 4: **Weakly supervised phrase grounding experiments on Flickr30K.** Adding additional cross-attention layers improves performance on the R@1 and pointing recall criteria. We include a description of the visual features and baseline approaches in the supplemental. IN and VG denote ImageNet and Visual Genome respectively.

| Approach | Visual Features | R@1 | R@5 | R@10 | Pt Recall |
|---|---|---|---|---|---|
| Fang *et al* [18] | VGG-cls (IN) | - | - | - | 29.00 |
| Akbari *et al* [3] | VGG-cls (IN) | - | - | - | 61.66 |
| Akbari *et al* [3] | PNAS Net (IN) | - | - | - | 69.19 |
| Align2Ground [16] | Faster-RCNN (VG) | - | - | - | 71.00 |
| Gupta *et al* [22] | Faster-RCNN (VG) | 51.67 | **77.69** | 83.25 | 76.74 |
| ours (word + sent): 1 CA layer | Faster-RCNN (VG) | 49.99 | 76.72 | **83.51** | 74.97 |
| ours (word + sent): 2 CA layers | Faster-RCNN (VG) | **53.80** | 76.69 | 82.28 | **76.78** |

**Results and analysis.** We report results in Table 4. As mentioned above, omitting the region value projection layer results in a performance drop, as evidenced by the results obtained by using a single cross-attention layer. Adding a second cross-attention layer generally helps to improve the localization accuracy and leads to notable increases in R@1 and pointing hand recall accuracies, surpassing state-of-the-art [22] on those criteria. We note that our full CoMMA model with unimodal self-attention layers hurts performance (R@1 = 48.81%). One possible reason is that optimizing CoMMA on this task is challenging and it may be overfitting. However, it is very promising that adding more layers works out of the box in general on a task that is typically addressed by shallow attention models. In contrast, we show in the supplemental that stacking multiple layers in our default CoMMA architecture when trained with the sentence-level loss $\mathcal{L}_{sent}$ only results in improved R@K localization accuracies.

Finally, we also observe that, unlike the observations in the video experiments, the word-level loss is critical for phrase grounding in images. We hypothesize that this finding is a result of the captions consisting primarily of noun phrases and being strongly correlated with the images. In contrast, the video task requires spatially localizing different interactions that are mentioned in a narration, which are more semantically complex as compared to noun phrases, making the word loss less effective.

**Discussion.** This work introduces the problem of localizing narrated interactions in uncurated videos without localization supervision. To address this task, we propose a novel Contrastive Multilayer Multimodal Attention module that facilitates repeated attention interactions between spatiotemporal regions and words to learn their latent alignment. Additionally, we introduce a new evaluation dataset that provides spatial annotations for narrated instructions in videos. Evaluations on the three separate tasks of interaction and object localization in videos as well as phrase grounding in images demonstrate the ability of our model to ground both words and phrases more effectively than the shallow and full-attention baselines. Our approach opens up the possibility of further exploration in other modalities. Learning to associate relevant spatial regions with natural language sentences may also be beneficial to learning richer video representations, particularly with the recent gravitation towards self-supervised representation learning.

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
