# A Supplementary

In this supplementary material, we provide the following additions to the main submission:

## A.1 Weakly supervised phrase grounding experiments

**Implementation details.** Following Gupta *et al.* [22], we extract image region features with a Faster-RCNN [43] model pretrained on Visual Genome [29] and contextualized word embeddings from a pretrained BERT [17] model. The text value projection is implemented as a two-layer MLP with Batch Normalization [25]. We use ReLU as the activation function. We set the joint embedding space dimension to the same as Gupta *et al.* [22] (d=384) and use a batch size of 50 images. We adopt a constant learning rate of 1e-5 and the Adam optimizer. Our approach is built on the publicly available implementation of Gupta *et al.* [22]. Finally, we train our model for 8 hours on a single V100 GPU.

**Hybrid model.** We provide an illustration of our weakly supervised phrase grounding model in Figure 4b (this supplemental). $D_w$ and $D_r$ denote the dimensions of the input word and region features while $N_w$ and $N_r$ denote the number of words and regions, respectively. We use $D$ to indicate the dimension of the joint embedding space. To incorporate our proposed CoMMA into the model of Gupta *et al.* [22] (Figure 4a (this supplemental)), we begin by removing a value projection MLP for the region features. This is due to the fact that our proposed CoMMA module does not project the region features to different subspaces for computing cross-attention. The model of Gupta *et al.* [22] is illustrated in Figure 4a (this supplemental). The outputs of the region and word key projection MLPs are passed as inputs into our cross-attention layer which computes contextualized region and word representations, respectively. We use the final contextualized word representations to compute the sentence-level loss. We found that using the contextualized word representation outputs from the first cross-attention layer to compute the word-level contrastive loss is critical for optimizing the model successfully. Finally, the sentence loss is weighted by a hyperparameter. We use the attention mask from the first cross-attention layer during evaluation.

**Negative Noun Loss.** In our experiments, we also adopt a negative noun loss as used in Gupta *et al.* [22]. Specifically, we create context-preserving negative captions for an image by substituting a noun in its original caption with negative nouns, that are sampled from a pretrained BERT [17] model. We compute the negative noun loss by contrasting the image region representations against these negative captions. For the rest of this supplementary, we use the term 'word-level contrastive loss' to include the negative noun loss as well.

**Ablation Results.** Table 5 (this supplemental) reports ablation results of the layers in our proposed CoMMA module when trained only with the sentence-level contrastive loss. In this case, adding cross-attention and self-attention layers generally helps to improve localization performance. As mentioned

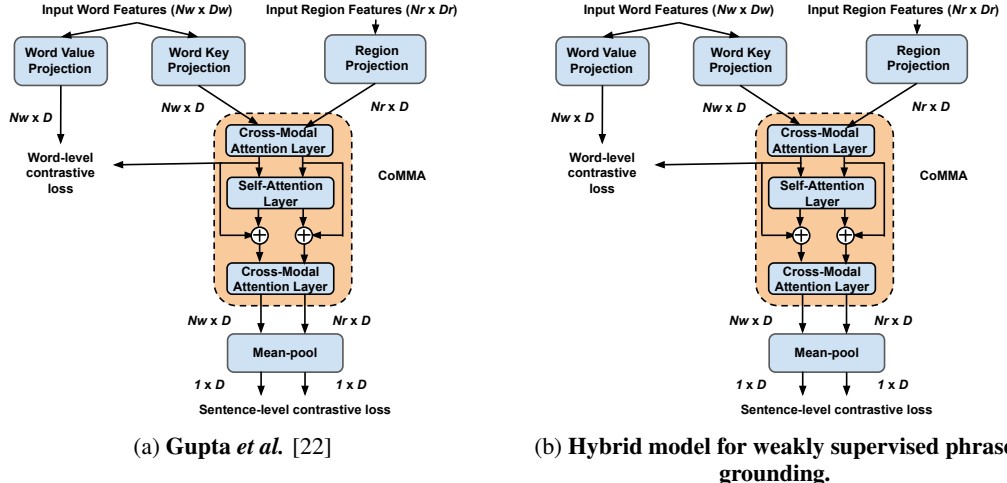

(a) **Gupta *et al.*** [22]    (b) **Hybrid model for weakly supervised phrase grounding.**

Figure 4: $N_w$, $N_r$, $D_w$, $D_r$ and $D$ denote the number of words, regions as well as the dimensions of the input word and region features and the joint embedding space, respectively.

Table 5: **CoMMA component ablation on Flickr30k with only sentence-level loss.** We observe that adding cross-attention and self-attention layers generally help to improve the localization accuracy across the Recall@K criteria.

| Cross-Modal Attention 1. | Cross-Modal Attention 2 | Self-Attention | R@1 | R@5 | R@10 | Pt Recall |
|---|---|---|---|---|---|---|
| ✓ | ✗ | ✗ | 35.43 | 64.48 | 73.92 | 68.68 |
| ✓ | ✓ | ✗ | 39.03 | 66.91 | 75.15 | **70.13** |
| ✓ | ✓ | ✓ | **39.31** | **68.43** | **77.35** | 67.36 |

above, we remove the value projection for region features from the model of Gupta *et al.* [22] to incorporate the CoMMA module. This modification results in an initial drop in performance when a single cross-modal attention layer is used, as shown in the first row of Table 6 (this supplemental) and Table 4 (main paper). We also observe that the sentence-level contrastive loss is complementary to the word-level contrastive loss (on its own, it performs reasonably well). For example, in the case where only one cross-attention layer is used, adding the sentence-level contrastive loss leads to a 2.5% in the R@1 accuracy. However, using the complete CoMMA model hurts performance. We hypothesize that it is overfitting on the smaller dataset and we have not found the optimal optimization setting. Finally, Table 7 (this supplemental) reports the ablation results over the different weight values of the sentence-level contrastive loss. We achieve the best performance when we use a weight value of $\lambda = 0.005$ with two cross-attention layers.

## A.2    Grounding narrated instructions in videos

**Self-training dataset** The HowTo100M [35] dataset contains approximately 1.2 million instructional videos that are sourced from YouTube, encompassing multiple domains such as cooking and hand-

Table 6: **Adding sentence loss to our hybrid model on Flickr30k.** We observe that the sentence loss is complementary to the word loss (on its own, it performs reasonably well). We also note that adding more attention layers also helps to improve localization performance.

| Cross-Model Attention 1 | Cross-Model Attention 2 | Self-Attention | Sentence Loss | R@1 | R@5 | R@10 | Pt Recall |
|---|---|---|---|---|---|---|---|
| ✓ | ✗ | ✗ | No | 49.99 | **76.72** | **83.51** | 74.97 |
| ✓ | ✗ | ✗ | Yes | 52.43 | 76.11 | 81.75 | 76.65 |
| ✓ | ✓ | ✗ | Yes | **53.80** | 76.69 | 82.28 | **76.78** |
| ✓ | ✓ | ✓ | Yes | 48.53 | 76.26 | 82.98 | 73.78 |

Table 7: **Sentence loss weight ablation with our hybrid model on Flickr30k.** The best performance is obtained with two cross-attention layers with the weight of the sentence loss set to 0.005. In the case where only one cross-attention layer is used, increasing the value of $\lambda$ from 0 to 0.01 or 0.005 leads to a 2% in the R@1 accuracy.

| Cross-Model Attention 1 | Cross-Modal Attention 2 | Self-Attention | Sentence Loss Weight $\lambda$ | R@1 | R@5 | R@10 | Pt Recall |
|:---:|:---:|:---:|:---:|:---:|:---:|:---:|:---:|
| ✓ | ✗ | ✗ | 0.0 | 49.99 | **76.72** | **83.51** | 74.97 |
| ✓ | ✗ | ✗ | 0.01 | 52.59 | 75.42 | 80.82 | 75.80 |
| ✓ | ✗ | ✗ | 0.005 | 52.43 | 76.11 | 81.75 | 76.65 |
| ✓ | ✗ | ✗ | 0.001 | 45.72 | 71.05 | 77.84 | 71.99 |
| ✓ | ✓ | ✗ | 0.01 | 52.44 | 75.55 | 81.13 | 75.71 |
| ✓ | ✓ | ✗ | 0.005 | **53.80** | 76.69 | 82.28 | **76.78** |
| ✓ | ✓ | ✗ | 0.01 | 46.23 | 73.32 | 80.70 | 72.94 |
| ✓ | ✓ | ✓ | 0.01 | 47.57 | 75.59 | 82.63 | 72.92 |
| ✓ | ✓ | ✓ | 0.005 | 48.53 | 76.26 | 82.98 | 73.78 |
| ✓ | ✓ | ✓ | 0.01 | 48.81 | 76.33 | 82.99 | 75.32 |

crafting. These videos contain transcribed narrations that are either uploaded manually by users or are the output of an automatic speech recognition (ASR) system. Furthermore, the narrations have been pre-processed to remove all stop words. As noted in prior work [35], there is a weak temporal alignment between the narrations and the video clips, adding another degree of difficulty in training with such videos. The misalignments are due to the nature of instructional videos where an interaction is often narrated before or after demonstrating it.

**Implementation details.** To avoid the time-consuming and computationally expensive process of pretraining the feature encoders used as inputs to our model, we leverage the publicly available pretrained weights of a S3DG model [35] trained with self-supervision. We use Word2vec embeddings [37] to represent semantic information contained by words in the narrations. Note that our proposed model is still fully self-supervised since the pretraining step for both the video and language encoders is also performed on the HowTo100M dataset without curated temporal or language annotations. The language encoder leverages pretrained Word2Vec [37] embeddings. In our experiments, we set the dimension of the joint embedding space $D$ to be 512. We use identity and linear projections with the same input and output dimensions to represent $W_K$, $W_Q$ and $W_V$ in cross-attention and self-attention layers, respectively. To train our proposed model, we set a learning rate of 1e-4 and optimize the model using the AdamW [32] optimizer. At the beginning of training, we apply linear warmup for one epoch. We train our model on 8 V100 GPUs which takes about a day. During inference, we adopt the practice in prior work where the resolution of the input frame is set to 224 x 224. Each input video clip consists of 16 frames. The dimensions of the final feature map are T x H x W x D = 2 x 4 x 4 x 512, where T, H, W, D correspond to the temporal, height, width, and feature dimensions.

**Mathematical formalization of the cross-attention layer** We provide a more detailed formulation of the cross-attention layer in our CoMMA module here. In particular, we explain how we prevent the mixing of features from different modalities in this layer. Let us first consider an example where we have two features from different modalities $x_1$ and $x_2$. Let $x$ be the concatenation of the two modalities' features $x = [x_1; x_2]$. We denote the operation in a linear layer as $Ax$ for a matrix $A$. Let us write $A = [A_1 \ A_2]$. Then, $Ax = A_1 x_1 + A_2 x_2$. In this case, the two modalities' features are fused since there is a summation operation between them.

Now, let us consider our setup where we have sets of "values" features $V_1$ and $V_2$ for two modalities. When full attention is used as in Figure 2c (bottom), the output of the softmax function in Equation (1) is a full weight matrix $A = [A_{11} \ A_{12}; A_{21} \ A_{22}]$. When multiplying with the value matrix $V = [V_1 \ V_2]$ in Equation (1), the output is $[V_1 A_{11} + V_2 A_{21} \ V_1 A_{21} + V_2 A_{22}]$. Note that in this case, the feature sets for the two modalities ($V_1$ and $V_2$) are summed together.

When cross-attention is used as in Figure 2c (top-left), the output of the softmax function in Equation (1) is the matrix $A = [0 \ A_{12}; A_{21} \ 0]$. When $A$ is multiplied by the value matrix $V$, the output is $[V_2 A_{21} \ V_1 A_{12}]$. Consequently, the features for the two modalities are not summed together. The attention interaction between the two modalities only occurs when computing the softmax weights

$A_{12}$ and $A_{21}$. We emphasize that these softmax weights are not features themselves, but instead are used for softly selecting features in the two modalities.

**Implementation details of baseline architectures.** Section 3.1 (main paper) describes existing state-of-the-art cross-modal attention modules. We provide additional implementation details of the full-attention baseline that uses a sentinel vector [49] and the shallow attention module [3]. The full-attention model differs from our proposed CoMMA module in two major ways. First, it computes attention weights over the entire multimodal sequence of words and clip features (full attention) instead of between modalities. Second, it uses a sentinel vector to aggregate information between visual and language features. The sentinel vector is a learnable parameter that has a similar purpose to the classification token that is commonly used in BERT language models [17]. Similar to its counterpart in the BERT model, the sentinel vector is prepended to the concatenated sequence of words and visual features and passed through a transformer to aggregate visual-semantic context in it.

The shallow attention module in Akbari *et al.*[3] computes a shallow cross-modal attention between words and convolutional features from different levels of an image encoder. The shallowness of its model stems from the fact that it only consists of a single layer of cross-modal attention between the language and visual features. In contrast, CoMMA comprises of alternating layers of cross-modal as well as self-attention layers. Note that this is different from the CBT model since it does not use a full attention module. As evidenced by our results, using a deeper cross-modal attention module improves the ability of our model to find the latent alignment between the language and visual modalities.

Finally, in Table 3 of the main paper, we note that we were unable to exactly reproduce the reported NAFAE results as the original base features were not available. Our best attempt with newer features was 4% lower than the reported bounding box detection accuracy. The final pointing accuracy obtained with the new features is 41.65%. Thus, we report the official higher bounding box detection accuracy of 46.95%. Our approach significantly outperforms this baseline.

**Active Hands experiment implementations and results** We compare our approach to a strongly supervised baseline that localizes *active hands* and interacted objects [46] in Table 8. We evaluate their pretrained model on YouCook2-Interactions under 3 settings:

1. taking the center pixel of the detected interacted object bounding box
2. taking the midpoint of the detected left and right hand bounding boxes
3. using a combination of the (1) and (2). In the case where no interacted objects are detected, we use (2) alone.

The high performance obtained by the pretrained Active Hands model is expected since hands are very often used in interactions, especially in the domain of instructional videos. However, they also suggest that simply detecting hands will likely not solve the proposed task. In particular, it will be challenging to resolve the ambiguity in scenes with multiple interactions involving different people that are occurring simultaneously. Additionally, this approach is not able to work well if there are erroneous detections of hands or objects or multiple pairs of hands visible. This is especially relevant in footage where the hands are small, making recognition based on local appearance difficult. Furthermore, detecting interacted objects using detected hands alone may not be feasible if the described interaction occurs between two objects without any human actors.

One of the main challenges with using the Active Hands model is that there are different heuristics that can be used to evaluate the localization accuracy, as reflected by the 3 settings described above. More importantly, this approach relies on large-scale bounding box annotations of hands during training and evaluation. In contrast, our proposed approach learns to localize such interactions from publicly available data without relying on such hand-annotated bounding boxes. Also, we would like to emphasize the generalizability of our proposed approach to the phrase grounding task in images that does not involve prominent human actors. We hypothesize that using an active hand detector will not generalize as well as our approach to this task given its reliance on finding noun phrases (instead of interactions) in images. Finally, we note that, in theory, it is possible to apply the CoMMA module on top of the hand and object bounding boxes to potentially improve the performance on the task in [46].

**Limitations.** One limitation is our method operates over coarse feature maps. Future work can aim to refine the localization predictions to compute precise masks for the interactions.

Table 8: **Interaction localization evaluation on YouCook2-Interactions.** We compare our approach against the pretrained strongly supervised Active Hands [46] model under 3 different settings.

| Approach | Localization Accuracy (%) |
|---|---|
| Setting 1 | 70.40 |
| Setting 2 | 61.47 |
| Setting 3 | 64.47 |
| CoMMA (Ours) | **55.80** |

## A.3 YouCook2-Interactions

Due to a lack of suitable datasets for localizing narrated interactions, we introduce an evaluation dataset, YouCook2-Interactions, that provides spatial bounding box annotations for interactions that are described by natural language sentences. Our dataset is built on the validation split of the YouCook2 dataset [65]. The original YouCook2 dataset contains around 2000 videos sourced from YouTube that cover approximately 89 recipes. It provides temporal annotations for segments and accompanying natural language descriptions. Each temporal segment is denoted by its start and end times and is described by a natural language description. Each description contains at least one interaction involving multiple objects (e.g., cut the chicken into cubes and add them to a pan).

### A.3.1 Annotations

Our annotation process is split into three stages, which we will describe below. In the first stage (Person filtering), we filter frames using an off-the-shelf person detector. Next (Frame relevant labeling), we label each filtered frame as 'relevant' or 'irrelevant'. Finally (Bounding box annotations), we annotate each relevant frame with a spatial bounding box. We use qualified workers from the Amazon Mechanical Turk platform for both frame labeling and bounding box annotations.

**Person filtering.** The primary focus of our dataset is on localizing narrated interactions. As such, frames which do not contain any instances of a person are removed to reduce the number of frames for labeling and annotation. To begin with, we extract all of the frames in the relevant segments from the validation split and run the YoloV3 object detector [42] on all of the frames to detect instances of humans. If a person is not detected in a given frame, it is removed from the labeling and annotation stages. Close-up frames that are visually relevant to narrated actions are not filtered out as long the object detector detects part of a person such as a hand.

**Frame relevance labeling.** One crucial characteristic of these narrations is that each specified interaction is only relevant to certain parts of the temporal segment. Unlike existing datasets which simply aim to track a referred object across frames, the specified interactions may not be visually relevant throughout the entire segment. Consequently, in the first stage, we ask workers to label if frames in a provided segment are relevant to any interactions specified within the description. We define people to be interacting with objects if they use their hands to perform an action involving the objects such as stirring or cutting. For each frame, three workers annotate whether the it is relevant and we select the majority label as the ground-truth label.

**Bounding box annotations.** After the set of relevant frames from each segment has been determined from the first stage of annotation, workers are tasked to draw a bounding box around the relevant region on a given frame that corresponds to the described interactions. We provide a general overview of the guiding principles here. See below for the entire set of principles. Since the primary objective of our approach is to localize interactions, the ideal bounding box annotation would enclose the region where the objects are being interacted with according to the natural language description. In cases where multiple objects are mentioned in the narration, in a given frame, only those that are being interacted with should be enclosed. Workers are also asked to enclose the entire hand and the object if the object is partially occluded in a frame.

The following sections include the guiding principles given to the annotators that are used in both the frame relevance labeling and bounding box annotation stages. For both stages, annotations are done on frames that are extracted at 1 frame per second (fps) and workers are asked to watch the entire clip to understand the context before they start annotating.

**Sentence: add the chicken cubes into the instant pot**

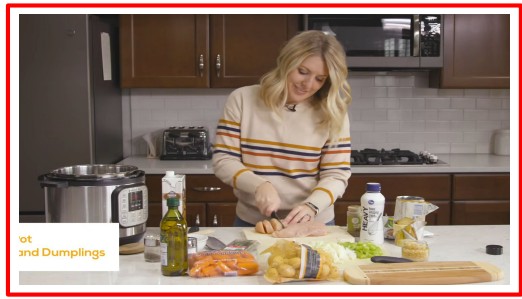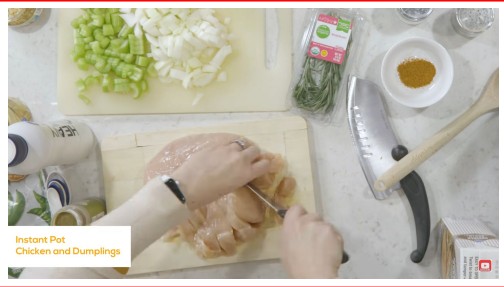

**Explanation: both frames are irrelevant because cutting the chicken is not specified in the description.**

Figure 5: Visual example of labeling frames as irrelevant because the sentence is not visually relevant. (Video credit: Six Sisters' Stuff [48])

### A.3.2 Frame Relevance Labeling Guidelines

In the original temporal segment annotations provided in the YouCook2 dataset, we observed that the proportion of relevant frames within the specified temporal segments that corresponds to the described interactions is relatively small. To ensure that only relevant frames have bounding box annotations, the first step in our annotation process tasks workers to watch a given video clip and label if the frames are relevant. In this task, we define people to be interacting with objects if they use their hands to perform an action involving the objects such as stirring or cutting. Please note that if a person is just looking or pointing at the object, it is *NOT* considered as an 'interaction'.

A frame is labeled as '**irrelevant**' if it fulfills one of the following conditions:

1. it depicts a scene where the person would describe the objects or ingredients being used but the specified actions are not being carried out.

2. some sentences and their interactions may be ambiguous (*e.g.*, boiling water, bake the pizza in the oven). Frames where the person does not interact with the specified objects should be labeled as 'irrelevant'.

3. if the action is observed but the objects being interacted with are not visible at all. However, if parts of the objects are visible, then frame should be labeled as 'relevant' (Figure 9) (this supplemental).

As noted, there are ambiguous actions like 'cooking chicken in a pan', 'boiling water' or 'baking pizza in an oven'. In such cases, all frames with interacted objects as the action is being carried out should be labeled 'relevant'. Frames where the person does not interact with the object should be labeled as 'irrelevant'. We provide several visual examples with reasoning for their labels in Figures 5, 6 and 7 (this supplemental). The frames that are outlined in red and green are marked as 'irrelevant' and 'relevant', respectively.

### A.3.3 Bounding Box Annotation Guidelines

Given the set of relevant frames that have been filtered by the first stage, Amazon Mechanical Turk workers are tasked to draw bounding boxes around the relevant regions given a natural language sentence. A visualization of the annotation interface can be found in Figure 8 (this supplemental). The general guiding principles are listed as follows:

1. Bounding boxes should only target the objects involved in the action as well as the entire hand / hands interacting with them. If only one hand is interacting with the object, the free hand should NOT be included.

2. The bounding box should only enclose the object(s) being interacted with, not all objects mentioned in the sentence.

**Sentence: turn the corn cobs a few times as they are cooked**

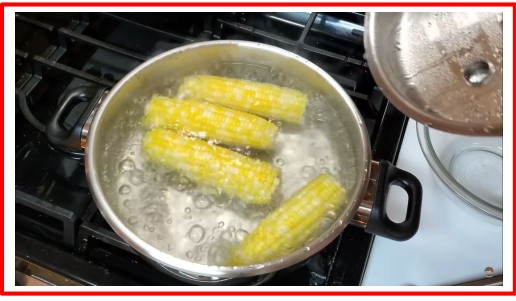 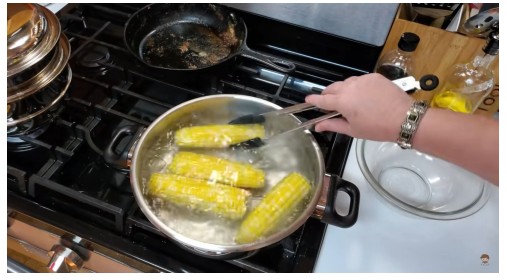

**Explanation: The frame outlined in green is relevant since the person is in the process of flipping the corn cobs.**

Figure 6: Visual example of labeling frames based on their visual relevance to the sentence. (Video credit: Collard Valley Cooks [13])

**Sentence: pour the quinoa into the pan**

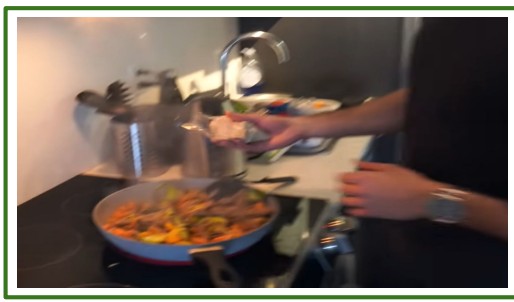 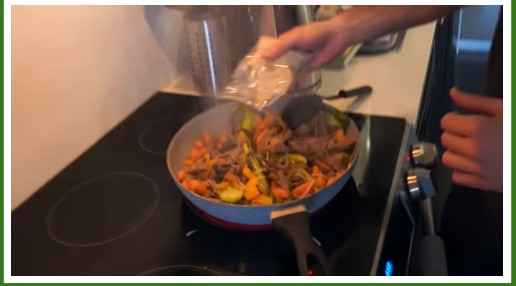

**Explanation: these frames are relevant since he is beginning the action of `pouring`.**

Figure 7: Visual example of labeling frames based on their visual relevance to the sentence. (Video credit: Kharma Medic [34])

3. It should be as tight as possible.

4. In cases where the specified object is partially hidden, the bounding box should still enclose the entire hand and the object. For example in Figure 9 (this supplemental), some parts of the object are occluded by her hand. However, the bounding box still includes the entire hand.

Note that the HowTo100M dataset contains videos that overlap with those of YouCook2. To maintain a proper evaluation setting, overlapping videos are removed from the HowTo100M dataset. In our dataset, an interaction is defined as an action that involves the manipulation of at least one object.

### A.4 Dataset statistics

We show dataset statistics in Table 9. In general, our dataset contains approximately 256 video segments that are split among 92 videos. Each video segment is denoted by its start and end times and corresponds to a single narration. The vocabulary encompasses 54 unique verbs and 73 object categories. Since the vocabulary is extracted from the YouCook2 dataset descriptions, they are mainly restricted to the culinary domain.

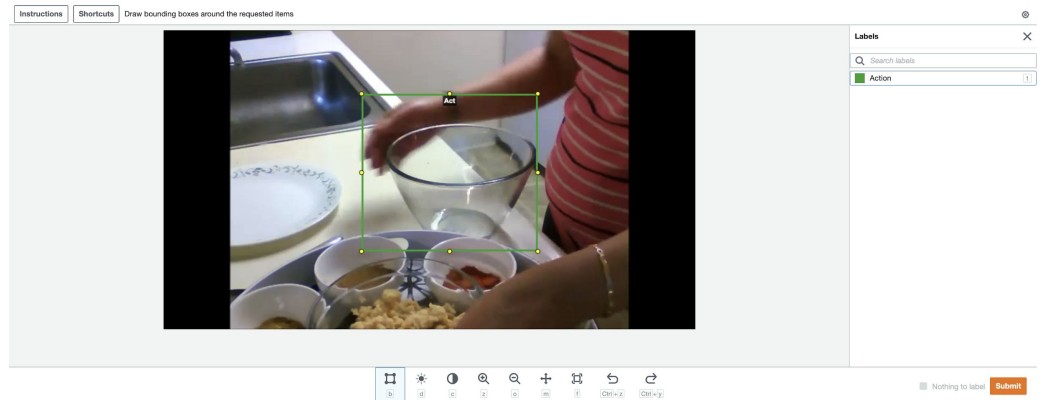

Figure 8: An example of the bounding box annotation interface used on Amazon Mechanical Turk. (Video credit: Curiosity Culture [14])

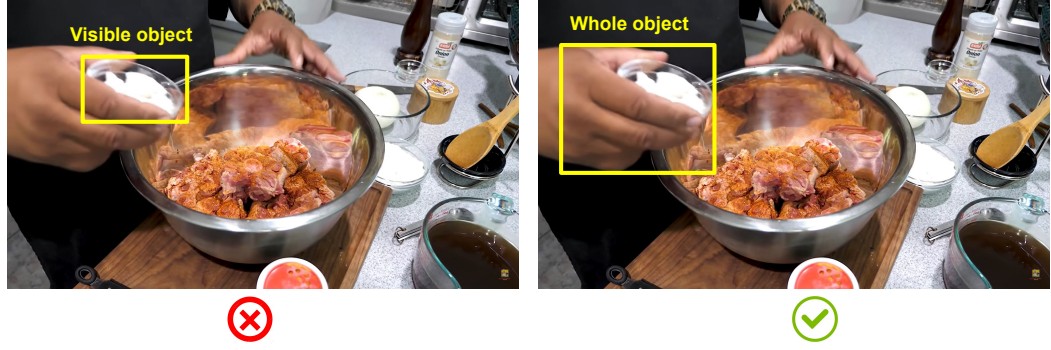

Figure 9: Visual example of how to annotate a bounding box when only part of the object is visible. (Video credit: Smokin' Grillin' wit AB [56])

| | |
|---|---|
| # segments | 256 |
| # videos | 92 |
| Average segment length | 24 sec. |
| # frames | 6238 |
| # verbs | 54 |
| # nouns | 73 |

Table 9: **YouCook2-Interactions dataset statistics.** Notice that our collected evaluation dataset with bounding box annotations supports a range of described actions and objects.