# OpenReview forum: "Look at What I’m Doing: Self-Supervised Spatial Grounding of Narrations in Instructional Videos"
_NeurIPS.cc/2021/Conference — NeurIPS 2021 Spotlight_

### Official Review · Reviewer_Broi · 2021-07-16

**Rating:** 7
**Confidence:** 4

**Summary:**

This paper concerns a new problem of grounding narrated interactions in video. It is similar to existing work on (object) phrase localization/grounding, but with a twist on grounding more complex narrations that could involve both entities and predicates (e.g., put tomatoes into the baking tray). The model training is performed in a weakly supervised fashion on a subset of HowTo100M. Training loss is the standard InfoNCE loss at the sentence level (vs. word-level as in existing grounding work). Model architecture is designed to capture inter-modal correspondence through cross-modal attention. Evaluation is conducted on YouCook2-BB, Flickr30k-Entities, and a newly collected dataset for narration grounding called YouCook2-Interaction.

**Limitations And Societal Impact:**

Appear to be.

**Main Review:**

The main contribution of the work comes from the proposed weakly-supervised grounding framework. It demonstrates that i) with a carefully-designed decoupled multimodal interaction layer (the bottom line of Tab. 2, right), grounding performance can be greatly enhanced, ii) and sentence-level contrastive loss has an edge over the traditional word-level contrastive loss. Experimental results and ablation studies are overall comprehensive.

In terms of the weaknesses of the work, the paper can be significantly benefited from better clarity, with main technical contributions highlighted. In the current shape, the paper attempts to motivate with the new problem setting (and dataset), which is somewhat incremental considering existing work and complicates the storyline. More detailed comments are as follows.

i) The motivation is somewhat disconnected from the method itself. For example, at L54, the paper mentioned that "[...] there are multiple 'onion' objects visible and the correct one depends on the action being applied." However, the paper has not addressed this problem particularly but instead proposes a rather generic approach for grounding. Also, the YouCook2-BB benchmark adopted in the experiment is agnostic to if the action is applied (i.e., 'onion' will be annotated throughout the video as long as it's visible and will be cut at some point).

ii) The pointing hand evaluation metric is somewhat weak. Consider proposal-based metrics. Also, in L257, what if there are multiple boxes match the same narration?

iii) The proposed method is based on a well-trained model (appear to be MIL-NCE? More details should be included in the main paper) and further refined on a well-curated dataset for the downstream task. It's not surprising that it performs well (which is somewhat unfair to methods without access to these resources). More analyses need to be conducted on the impact of each factor.

iv) Minor comments:

L227, should "for the j-th word" be "region" instead?

L230, "self-training" should be "pre-training". "Self-training" usually refers to a training scheme that leverages generated pseudo labels.

L277, state-of-the-art.

**Time Spent Reviewing:**

3

---

> ### Author Response · Authors · 2021-08-11
> **Response to reviewer Broi**
>
> **The paper can be significantly benefited from better clarity, with main technical contributions highlighted.**
>
> Thank you for your review. Please look at the general response where we highlight the main technical contributions of this work. We are happy to incorporate additional feedback to improve clarity.
>
> **The paper proposes a generic approach for grounding. In addition, the YouCook2-BB benchmark adopted in the experiment is action-agnostic.**
>
> Thank you for your comment. We agree that the strength of our approach lies in its generalizability across the two demonstrated separate tasks of spatially localizing described interactions in videos as well as phrase grounding in images. Given the model-agnostic nature of our proposed CoMMA module, it can also be easily integrated with any base encoders on different tasks to improve performance on downstream tasks. We will emphasize this point more in the introduction. Note that we also want to call out the new interaction localization task in videos since it is novel and not well-studied in prior work.
>
> We agree that the YouCook2-BB benchmark is agnostic to whether the action is applied. We evaluated our pretrained model on the YouCook2-BB dataset to determine its generalizability to a slightly different task of localizing single object word queries instead of longer text sequences. Based on the observed results, our model has shown to be relatively effective at both tasks (spatially localizing interactions and objects).
>
> **Consider proposal-based metrics. How to handle multiple overlapping bounding boxes (L257)?**
>
> We note that the pointing hand metric is one of the standard criteria used in evaluating image grounding models [3, 17]. If requested, we are happy to try to incorporate unsupervised object proposals generated by selective search into the proposed approach for the camera-ready version.
>
> The YouCook2-BB dataset has overlapping bounding boxes for different objects. However, we adopt the same practice as in prior work [36]  by evaluating our model with individual object word queries. In our YouCook2-Interactions dataset, there is only one bounding box for each described interaction.
>
> **Include more details on the MIL-NCE model and how it is used; more analyses need to be conducted on the impact of each factor.**
>
> Thank you for raising these points. We will include a more detailed analysis in the main paper on the impact of additional factors such as batch size and the types of encoders used on the final performance. We first note that our approach is model-agnostic and can be integrated with any pretrained visual and text encoders, where they are kept fixed during training.  To avoid time-consuming and large-scale training from scratch, we build on the pretrained MIL-NCE model as it has a strong text-to-video retrieval accuracy. Our video model based on MIL-NCE that includes our CoMMA module can be trained on 8 V100 GPUs in approximately 8 hours. When applying our CoMMA module with the weakly supervised phrase grounding model [17], the entire model can be trained in 8 hours on a single V100 GPU.
>
> For comparison, we implemented another baseline using the pretrained MIL-NCE without our CoMMA module, as suggested by reviewer rFC4. Specifically, a spatiotemporal grid of features is obtained by removing the global average-pooling layer from the visual encoder. We compute a matrix multiplication between the spatiotemporal features and the sentence feature to obtain a similar attention map output for inference. This method obtained a localization accuracy of 27.18%. This suggests that a model trained to align video and text representations for retrieval may not be focusing on the relevant regions during training and evaluation.
>
> We are happy to include additional suggested analyses for the final version.

---

### Official Review · Reviewer_gTL6 · 2021-07-17

**Rating:** 6
**Confidence:** 4

**Summary:**

The paper presents an approach for self-supervised spatial localization of interactions in videos by leveraging a large corpus of video clips and narrations. An alternating attention mechanism is presented that combines cross-modal and self (within modality) attention schemes. Pre-trained models on the HowTo100M dataset are evaluated by using pointing hand accuracy on a subset of the YouCook2 dataset, interaction annotations are also created in this work. The approach is also applied to grounding in Flickr30k images.


**Limitations And Societal Impact:**

The authors claim that they will release code judiciously to prevent "further development of surveillance and tracking technology". However, it's not clear how this will be executed.

**Main Review:**

*Strengths:*
1. Interesting work on leveraging the power of large datasets in a consistent domain (here, cooking related videos) to help learn spatial localization.
2. Fig. 2 explains the proposed approach in a very clear way. In fact, I would recommend reducing the size of Section 3.1 since the figure is very clear and Transformers have made this type of attention common and instead focus on Attention Rollout used during inference + additional experiments / clarifications. The current explanations in 3.1 are also causing some confusion (see below).

*Weaknesses:*
1. Claims:
a) Lines 118-119 indicate that the proposed task is different from object grounding as it requires localizing interactions. However, I'm not sure why the proposed model would be good at localizing specific interactions such as "add onions to tray" and not just co-localize the objects "onions" and "tray".
b) Fig. 1 (right), seems to indicate that there are multiple instructions that are localized in various spatio-temporal parts of the video. However, none of the qualitative examples highlight the textual part. In fact, the rest of the paper seems primarily interested in spatial localization, not temporal. This is quite confusing and misleading.
c) I'm not sure why the paper claims "early interactions" (L46, L132), or "deep cross-modal attention" (L137) when as far as I understand, the module from Fig. 2c is applied to the final layer of features extracted from a CNN. See weakness 2c below related to this.

2. Models and ablations:
a) Word-level loss, Eq. 6. Firstly why is this used? Second, the paragraph talks about "aims to learn an alignment between each word in the narration and all spatial regions". However, the attention is computed between $S_j$ and $S_{fj}$ instead of $C_{fix}$. Do the authors wish to indicate that $S_{fj}$ through multiple layers of cross-modal attention already captures information in the spatial regions, or is this a typo?
b) Based on Fig. 2c, the extracted features go through a cross-modal, self-attention, and another cross-modal block. In my understanding, all this put together is one CoMMA layer. Eq. 2 and 3 seem like intermediate stages of one CoMMA layer should not use the same notation $Y_{i+1}$.
c) L198 states that "CoMMA allows for repeated early interactions without mixing of features from different modalities", but the first attention block in Fig. 2c is in fact a cross-modal attention. How is this explained?
d) Would stacking CoMMA layers on top of each other help (alternating cross and self attention)?


3. Experiments:
a) While the center prior is a good simple baseline, based on the qualitative examples, I wonder how well would a method that simply predicts the center of the "active" hand perform. For example, [A] can provide localization of the interacting object and hand quite well, and it seemed like most GT boxes were inside this region.
b) Alternatively, to truly justify the model's capability of learning spatiotemporal attention, it might be interesting to try it on videos which don't have human actors as the prominent agent.
c) This work seems to be quite related to the study of referring expressions in videos (VID-Sentence [B], Lingual OTB [21]). These datasets could act as other means to evaluate the proposed approach.
d) Results on Flickr30k are not coherent with YouCook2-Interactions, adding to the confusion of the contribution. The whole message changes here, as using self-attention hurts the performance quite drastically (48.8% R@1 with CoMMA block, 53.8% without self-attention in CoMMA block).
e) Minor: Table 2, left/right descriptions need to be interchanged (L296-305).

[A] Shan, et al. Understanding Human Hands in Contact at Internet Scale. CVPR 2020.
[B] Chen, et al. Weakly-supervised spatio-temporally grounding natural sentence in video. ACL 2019.


### Post rebuttal
**TL:DR; I've increased my rating to 6 (from 4) with some strong conditions**

I thank the authors for their detailed response. Especially appreciate the clarifications on several aspects of the submission, in particular, resolving the superscripts with reviewer rFC4. That would make it much easier to read the paper.

I also really appreciate the additional effort taken in evaluating hand detection models from [A] on the proposed dataset, and including a thorough evaluation of three settings. I agree with the authors that this can be thought of as a supervised method albeit, important to note that it is trained on a proxy task of detecting hands and objects and not actual interactions indicated in the text.

While the rebuttal claims that the proposed approach would learn interactions despite the presence of human actors, I have my doubts regarding this, especially when so much of the data points to looking at the hands. It would be really nice to include some qualitative examples where the GT is not close to the hands, and perhaps even a quantitative split by finding samples where hands are not detected. In a future work, I would also encourage the authors to look at using data beyond the human interactions typically present in HowTo100M (think documentaries), where interaction localization is much more tricky and non-obvious.

While the rebuttal is very promising, there are a lot of small and big issues to be improved in the submission before it can be accepted. The very fact that such a detailed rebuttal was necessary in the first place is a little concerning. I'm happy to increase my rating to 6, conditional on all the changes being incorporated, but cannot champion it in the current version. One additional round of review may not hurt much.

**Time Spent Reviewing:**

4

---

> ### Author Response · Authors · 2021-08-11
> **Response to reviewer gTL6**
>
> **Focus on Attention Rollout used during inference and clarifications on the experiments.**
>
> Thank you. We will incorporate these changes in the camera ready.
>
> **Why would the proposed model be good at localizing specific interactions instead of just co-localizing the objects mentioned?**
>
> Thank you for pointing this out. We will highlight the main differences between our proposed task and that of object grounding in the paper.  For example, one main difference is that we only want to localize the objects being used in the relevant interaction at any point in time. It is not sufficient to simply co-localize all mentioned objects since some of them are not used in the interactions at all.
>
> Consider an example narration “slice a tomato into pieces”. In this case, we only want the model to localize the tomato that is being used in the cutting action and not the rest of the tomatoes on the counter-top, since they will not be relevant to the action.  To solve this task, we hypothesize that a model has to be able to reason about irrelevant regions for each interaction described in a narration.  By computing contextual visual and text features via cross-modal and self-attention over words and spatiotemporal features, our model has the opportunity to learn to disregard objects that are mentioned but not being used in a specific interaction.
>
> **Figure 1 (right) is confusing and misleading.**
>
> Thank you for the suggestion, and sorry for the confusion. We will update Figure 1 to better reflect the capability of our model and include a teaser of the obtained results in the form of qualitative visualizations.
>
> **Why are the interactions claimed to be early when the module is applied to the final layer of features extracted from a CNN?**
>
> Thank you for pointing this out. We will clarify these parts in the next iteration of the paper.  Specifically, we will replace “early interactions / fusion” to “interactions” on L46 and L132. Additionally, we will replace “deep cross-modal attention” with “cross-modal attention model with multiple stacked layers”.  Note that, in light of these clarifications, we want to make a clear distinction between our strategy compared with a state-of-the-art text-to-video alignment model [25]. This latter approach [25] utilizes a dual encoder architecture where the similarity between the video and query text is computed as a dot product between the aggregated outputs of the visual and text encoders. In our work, we seek to go beyond the simple dot-product operation to include hierarchical cross-modal interactions between all words in the text and spatiotemporal regions in the video clip.
>
> **Why is the word-level loss used? Clarification of Eq (6).**
>
> Thank you for bringing this up.  State-of-the-art weakly-supervised language grounding approaches [3, 17] often use a word-level loss to learn a fine-grained alignment between every word and region. We trained a variant of our model using the same loss to directly compare against these prior approaches.
>
> Please look at the general response for our clarification on Eq 6.
>
> **Eq. 2 and 3 seem like intermediate stages of one CoMMA layer should not use the same notation $Y_{i+1}$.**
>
> Thank you for your feedback. We will update this in the paper and denote the outputs of the cross-modal and self-attention layers as $Y_{i+1}^{ca1}$,  $Y_{i+1}^{ca2}$ and $ Y_{i+1}^{sa}$, respectively.
>
> **How does the first cross-modal attention layer in the CoMMA module in Fig. 2c allow for repeated early interactions without mixing the different modalities’ features?**
>
> We define the mixing of features from different modalities when an element-wise sum is applied to features across the different modalities.  In this case, our proposed cross-attention layer allows both modalities to attend to each other without leaking any information between them.
>
> More formally, let us consider a simple example where you have two feature vectors from different modalities, $x_1$ and $x_2$. Let x be the concatenation of the two modalities' features $x = [x_1; x_2]$. In a linear layer, you have the operation $Ax$ for a matrix $A$. Let us write $A = [A_1 \ A_2]$. Then $Ax = A_1 x_1 + A_2 x_2$. We say that the two modalities' features are fused when there is a summation operation between them.
>
> Now let us consider our setup where we have sets of “value” features $V_1$ and $V_2$ for two modalities. When we use full attention (Fig 2a bottom), the output of the softmax in Eq (1) is a full weight matrix $A = [A_{11} \ A_{12} \; A_{21}  \ A_{22}]$. When multiplying with the value matrix $V = [V_1 \ V_2]$ (Eq (1)), we get $[V_1 A_{11} + V_2 A_{21} \ V_1 A_{21} + V_2 A_{22}]$. Note that the feature sets for the two modalities ($V_1$ and $V_2$) are summed together.
>
> When we use cross attention (Fig 2a top-left), the output of the softmax in Eq (1) is the matrix $A = [0 \ A_{12}; A_{21} \ 0]$. When multiplying by the value matrix $V$, we get $[V_2 A_{21} \ V_1 A_{12}]$. Therefore, the features for the two modalities are not summed together. The interaction between the two modalities only occurs when computing the softmax weights $A_{12}$ and $A_{21}$. Note that these softmax weights are not features themselves, but instead are used for softly selecting features in the two modalities.
>
> We will include this explanation in the main paper.
>
> **Will using multiple CoMMA modules help?**
>
> Thank you for this suggestion. Please look at our general response on using more than one CoMMA module.
>
> **How well would a method that simply predicts the center of the "active" hand [A] perform?**
>
> Thank you for suggesting this experiment. [A] is a strong fully supervised baseline for localizing objects interacting with hands (the model in [A] is supervised with bounding boxes around hands and interacted objects). This baseline is closely related to our task and serves as a good target for future work. Note we view this baseline as similar to how self-supervised learning techniques (e.g., MoCo, SimCLR) compare against fully supervised image classification baselines on ImageNet. By training on a large image dataset annotated with bounding boxes and labels, [A] learns to identify the active hands and interacted objects in each image.  We evaluated their pretrained model on our dataset under 3 settings:
> 1. Taking the center pixel of the detected interacted object bounding box
> 2. Taking the midpoint of the detected left and right hand bounding boxes
> 3. Using a combination of the (1) and (2). In the case where no objects are detected to be interacted with, we use (2) alone.
>
> The obtained results are listed as below:
>
> | Method      | Localization Accuracy (%) |
> | ----------- | :----: |
> | Setting 1  | 70.40   |
> | Setting 2  | 61.47      |
> | Setting 3  | 64.47   |
> | CoMMA  | 55.80   |
>
> These results are somewhat expected since hands are very often used in interactions, especially in the domain of instructional videos. Interestingly, our method appears to automatically learn without supervision that the hands are helpful in localizing interactions.  However, as evidenced by this baseline, simply detecting hands will likely not solve the proposed task, despite its strongly supervised nature.
>
> One of the main challenges with using this baseline is that there are different heuristics that can be used to evaluate the localization accuracy, as reflected by the 3 settings described above. More importantly, this approach relies on large-scale bounding box annotations of hands during training and evaluation. In contrast, our proposed approach learns to localize such interactions from publicly available data without relying on such hand-annotated bounding boxes. Additionally, this approach is not able to work well if there are erroneous detections of hands or objects or multiple pairs of hands visible. This is especially relevant in footage where the hands are small, making recognition based on local appearance difficult.  Furthermore, detecting interacted objects using detected hands alone may not be feasible if the described interaction occurs between two objects without any human actors. Also, we would like to emphasize the generalizability of our proposed approach to the phrase grounding task in images that does not involve prominent human actors. We hypothesize that using an active hand detector will not generalize as well as our approach to this task given its reliance on finding noun phrases (instead of interactions) in images. Finally, we note that, in theory, it is possible to apply our proposed module on top of the hand and object bounding boxes to potentially improve the performance on the task in [A].
>
> **It might be interesting to try it on videos which don't have human actors as the prominent agent.**
>
> Thank you for this great suggestion.  We will mention it in the paper for future work. We would like to emphasize our work depends heavily on the pretraining instructional videos dataset. In this case, the HowTo100M video dataset primarily consists of human interactions. As shown by the zero-shot evaluation results in [25], it is hard for existing  models to generalize to videos that have a large visual domain gap. Finally,  we note that narrated interactions in video are most common with humans.
>
> **Results on Flickr30k are not coherent with YouCook2-Interactions.**
>
> We apologize for not explaining this better.  We show in Table 5 of the supplementary material that adding the self-attention layer improves the bounding box recall accuracies but hurts the pointing-hand accuracy, when only the sentence loss is used. This is consistent with our observations for the video experiments. Interestingly, combining the word loss with the sentence level loss always results in a performance drop with the inclusion of a self-attention layer, even on videos. We will add these points to the paper.

---

> > ### Author Response · Authors · 2021-08-11
> > **Additional response**
> >
> > **VID-Sentence [B] and Lingual OTB [21] could act as other means to evaluate the proposed approach.**
> >
> > Thank you for suggesting these datasets. As mentioned in L113, these datasets focus on object grounding through time and do not include any interactions.
> >
> > **Table 2, left/right descriptions need to be interchanged (L296-305).**
> >
> > We will update Table 2 to reflect these changes in the next version of the paper.
> >
> > **It is not clear how the code can be released judiciously.**
> >
> > We apologize for the ambiguity in our statement. Specifically, we promise to include a disclaimer in the code release that states the use of this code for surveillance or tracking purposes is out of scope.

---

### Official Review · Reviewer_q3gz · 2021-07-20

**Rating:** 8
**Confidence:** 5

**Summary:**

- This paper considers the task of spatially grounding the language description in videos, specifically, instructional videos are used to train the model, where the language description and visual signals are likely to be synchronised.

- The proposed architecture consists of stacks of intra- and inter attention modules, which allows the video and language to attend each other at early stage, and effectively highlighting or suppressing the corresponding features.

- An evaluation dataset is proposed with bounding box annotations for interactions described by natural language sentences, and proposed approach shows good performance on it. In addition, authors have also evaluated on Flickr30K under the setting of weakly supervised language grounding, showing state-of-the-art performance.

**Ethics Review Area:**

["I don’t know"]

**Limitations And Societal Impact:**

Yes

**Main Review:**

Originality:

- The proposed problem in this paper is very challenging, as it aims to train models with the raw instructional videos without manual box annotations.

- The proposed architecture and training loss are adopted from other papers in the literature, thus, I would say there is no novelty on the computational module and loss function, however, I do think these elements are used in a good way here, and aiming to tackle and challenging problem.

Quality:

- The paper is of good quality, above the acceptance threshold of NeurIPS.

Clarity:

- The paper is well-written, easy to follow. One suggestion would be, try to clarify the Eq. 6 a bit more.

- In terms of the related work, the section on self-supervised learning is not satisfactory, since this paper is on grounding actions in video, I would suggest the authors to include more recent papers on self-supervised video representation learning, there is a rich line of work on this topic, while the authors only mention the pace prediction work.

Significance:

- The proposed idea can be potentially useful for visual grounding, visual question answering, and even generic object detector, thus I think it is a significant work.

**Time Spent Reviewing:**

3 hours

---

> ### Author Response · Authors · 2021-08-11
> **Response to reviewer q3gz**
>
> **Include more recent papers on self-supervised video representation learning**
>
> Thank you for your suggestion. We will include a more comprehensive review on state-of-the-art self-supervised video representation learning approaches and will cite the papers below. We are happy to include any other relevant citations.
>
> 1. MemDPC (https://arxiv.org/pdf/2008.01065.pdf)
> 2. https://openaccess.thecvf.com/content_cvpr_2017/papers/Fernando_Self-Supervised_Video_Representation_CVPR_2017_paper.pdf
> 3. Space-Time Correspondence as a Contrastive Random Walk
> 4. Speednet: Learning the speediness in videos
> 5. Learning and Using the Arrow of Time
>
> **Computation module and loss function novelty**
>
> Please look at the general response on paper novelty and contribution.
>
> **Clarification on Equation 6.**
>
> Thank you for your constructive feedback. Please look at the general response for our clarification on Eq 6.

---

### Official Review · Reviewer_rFC4 · 2021-07-21

**Rating:** 7
**Confidence:** 5

**Summary:**

The paper proposes a method for grounding sentences in videos and images without explicit localization supervision.  The approach rely on a cross-attention architecture that is used in a way to make contrastive learning possible (since it prevents the final feature from containing information from the other modalities while still enabling information to be shared across modalities earlier in the network). Localization can be obtained from the cross-attention patterns between the language and the visual signal.

The authors apply their method on a subset of the HowTo100M dataset for videos and evaluate on the YouCookBB object for objects and a newly introduced YouCookInteraction where they show improvements over baselines. The method is also applied on the Flickr30K dataset on images.

**Ethical Concerns:**

None.

**Limitations And Societal Impact:**

Yes.

**Main Review:**

## Strengths

- The paper tackles an important problem (localization without strong supervision)
- The proposed architecture is interesting as it enables to have early fusion of text and visual features while still allowing contrastive learning to be used.
- The quantitative numbers are good

## Weaknesses

- **Important technical clarifications**: there are a few key questions that needs clarifications as its not completely clear from the paper.
  - *Number of layers?* When talking about layers, if I understand correctly you refer to whether or not there is one or two CA block in the CoMMA layer? Have you also explored using more than a single CoMMA blocks?
  -  *Resolution of input videos and feature maps*: What is the size of the input videos in practice and the final feature map?
  - *Word level loss*: The word level loss description and equation are confusing. The text says `between each word in the narration and all spatial regions`. However the word level loss only contains features related to text `S^T_{fj}`. Is that because these features are extracted after the first cross attention layer and hence effectively contains visual features? This is not clear at all from the description. Figure (4) (b) was helping to understand this in suppmat but I am unsure this is the same architecture used for videos.
  - CoMMa architecture figure: why is there a + sign at the top of the figure? If I understand correctly it is important to not have residual connection to avoid information being leaked from one modality to another which would make contrastive learning trivial.
  - CBT baseline: When using full self attention, isn't there the problem mentioned in L42-48 as it allows final features to have both image and video information hence making the contrastive task trivial to solve? How come this does not affect that much the performance? This require a more precise description as it is currently not clear how this baseline can really work.
  - Equations (5): there is no dependence in $i$ in the loss. Some index are probably missing? Similar comment applies to equation (6).

- **Conclusion between video and image results do not seem to align**: It seems that the word loss is very important for the flick30K experiment but is not working for the video case. Similarly the number of cross attention layers seem to be optimal at 2 for videos while for images it depends on the metrics. Why does this discrepancy exist?

- **Additional baseline request without cross attention mechanism**: for the video model it would be interesting to compare to the localization ability of the S3D-G pretrained model you use from [24]. For that a simple modification can be done by simply removing the average pooling [here](https://github.com/antoine77340/MIL-NCE_HowTo100M/blob/master/s3dg.py#L342) hence obtaining a spatio-temporal grid of features similar to the one used in the work. Then the fully connected layer can simply be applied on the grid of features to obtain a grid of feature living in the joint text-video space (see [here](https://github.com/antoine77340/MIL-NCE_HowTo100M/blob/master/s3dg.py#L345)). This will result in a matrix of visual features of shape `V= [X, d]` where `X` is the number of spatio temporal features and `d` is the dimension of the joint text-video space. Finally to obtain the scores for each feature in the grid given a text query, one can simply do a matrix multiplication with the text vector of the query `y` of shape `[1,d]` by doing `np.matmul(V, y.T)`. Then the same evaluation method can be applied. This would give another interesting point of comparison.

- **Minor comments**:
  - L296-L297: Table (left) should be Table (right),
  - nit the Tables could be made more pretty, e.g. following guidance from [here](https://people.inf.ethz.ch/markusp/teaching/guides/guide-tables.pdf)
  - Typo: inclided L328

## Overall assessment

The paper presents an interesting approach for an important problem. However some important things require additional clarifications. For this reason I am putting a borderline score for now (slightly leaning towards reject but I can update my score based on the rebuttal).

## Post reading reviews and responses

I read the other reviews and responses.

## Post Rebuttal comment

I would like to thank the authors for their effort in the rebuttal. I very much agree with what Reviewer gTL6 said, i.e. in the current version there are lots of things to incorporate for the camera ready to be really good.

That being said, given the engagement of the authors in the discussion, especially the clarifying discussion about the notations, I am ready to raise my score to accept (7) and be optimistic that the authors can include all the recommendations reviewers made to the final version of the paper. In particular I found the idea of cross attention quite clever and it also seems to work well (but it has to be clearly presented for it to be understood and impactful in the community), hence my leap of faith in the authors for this paper.



**Time Spent Reviewing:**

5

---

> ### Author Response · Authors · 2021-08-11
> **Response to reviewer rFC4**
>
> **Number of layers**
>
> In this work, we define a layer as either a cross-modal attention or self-attention operation. As such, each CoMMA module consists of three attention layers (cross-attention => self-attention => cross-attention), as depicted in Fig 2(c) of the main paper. Please also look at our general response on using more than one CoMMA module.
>
> **Resolution of input videos and feature maps**
>
> We apologize for leaving these details out and we will include them in the camera ready. During inference, we adopt the practice in prior work where the resolution of the input frame is set to 224 x 224. Each input video clip consists of 16 frames. The dimensions of the final feature map are T x H x W x D = 2 x 4 x 4 x 512, where T, H, W, D correspond to the temporal, height, width, and feature dimensions.
>
> **The word level loss description and equation are confusing.**
>
> Please look at the general response for our clarification on the word-level loss.
>
> **CoMMa architecture figure: why is there a + sign at the top of the figure?**
>
> Thank you for pointing this out. The plus signs at the top of Fig 2(c) are typos. We apologize for this oversight and will remove the plus signs at the top of the figure.
>
> **Why does the full attention mechanism in the CBT baseline not cause the performance to drop  by a huge margin?**
>
> Thank you for the feedback.  Indeed, we observed that the CBT baseline may be prone to poor solutions. As reported in Table 1 of the main paper, the CBT baseline (Full attention + sentinel) achieves an accuracy of 39.03%, which is close to the center prior, while our proposed approach outperforms it by about 15%. Quoting [35], we note that the CBT baseline network outputs a “Mutual Information (MI) -like score” that is plugged in directly to the NCE loss; it does not compute a cosine similarity score between the video clip and natural language query features.  We hypothesize that computing this MI-like score (instead of directly contrasting the two modalities’ features) may somewhat mitigate the CBT baseline from catastrophic failure. We will include visualizations of the CBT baseline in the paper and provide a more detailed analysis on this discrepancy in the camera ready.
>
> **There is no dependence on the index “i” in the loss formulations in Equation 5 and 6.**
>
> Thank you for pointing this out. We will include the dependence on index “i” in Loss (5) and (6) in the next version. In Equations (5) and (6), “n” refers to the total number of training samples within a sample batch and “i” indicates the specific pair of video clip and narration for which the loss is computed.
>
> **Why does the word-level loss seem to be important for the Flickr30K experiments but not for the video experiments?**
>
> The motivation for using the word-level loss is that it is good for localizing noun phrases as shown in [3, 17]. This loss aligns well with the Flickr30K phrase grounding task since the query captions are strongly related to the images and primarily consist of noun phrases. In contrast, the video task requires spatially localizing different interactions that are mentioned in a narration, which are more semantically complex as compared to noun phrases, making the word loss less effective. Moreover, the text annotations in the video training dataset have a weak temporal alignment with the associated video clips (L288-291). Consequently, certain words that are present in a narration may not be visually relevant at all in the video clip. This is the primary motivation behind the MIL-NCE approach used in [25], which aims to select correct narrations from a set of candidate narrations that are temporally near the video clip. As such, trying to align each word to a region in the word-level loss may be counterproductive since some words may not correspond to any region at all.
>
> Regarding the number of cross-attention layers, one consistent observation is that using an additional cross-attention layer improves performance on the most important evaluation criteria of localization and R@1 accuracies by a healthy margin on both the video and image tasks, respectively (Tables 2 and 4). In addition, the resulting R@5 accuracy for the image phrase grounding task is on par with that of [17], albeit with a slight degradation for R@10 accuracy.
>
> We will clarify both points in the paper.
>
> **Additional baseline request without cross attention mechanism**
>
> Thank you for suggesting this baseline. We evaluated this baseline on our dataset and obtained a localization accuracy of 27.18%. The huge difference between the aforementioned result and the performance achieved by our model of 55.80% suggests that a model trained for retrieval may not be focusing on the relevant regions as described by the natural language query. We will include this result in the paper.

---

### Author Response · Authors · 2021-08-11
**General response**

We really appreciate and thank all reviewers for their insightful and constructive feedback on our submission. We would like to emphasize the main technical contributions in this work, which are listed as follows:

1. We introduce the task of learning to spatially localize narrated interactions from instructional videos without relying on the spatial location supervision during training time. This task helps to circumvent the time-consuming and expensive process of annotating bounding boxes for training.

2. Our proposed approach incorporates alternating cross and self-attention layers without aggregating features across the video and language modalities. We show that this design is especially crucial to prevent the model from learning a trivial solution when the model directly compares contextual features from two modalities with a cosine similarity via a contrastive loss during training.

3. We introduce an evaluation dataset of bounding box annotations for described interactions in videos (YouCook2-Interactions).

We are happy that Reviewers rFC4, q3gz, and gTL6 thought that the task of localization without strong supervision via bounding box annotations is a very important and novel task. In addition, reviewer gTL6 noted that our work is interesting due to the fact that it leverages a large dataset to learn these localizations. Reviewer q3gz also noted that our approach builds on findings from current attention models and the contrastive loss in a good way to address this novel and challenging problem. Finally, reviewer Broi also noted that our proposed CoMMA module learns to address this new problem of grounding complex narrations through its decoupled multimodal attention layers in the self-supervised setting.

In the remainder of this response, we address the common questions on the word-level loss in Equation 6 as well as using multiple CoMMA modules here before responding to each reviewer individually on the remaining comments.

**Equation (6) - Reviewers rFC4, q3gz and gTL6**: Our apologies for the confusion on this loss. We further clarify it here. Following the notation in our paper, let $S_k$ and $C_k$ denote matrices of contextualized features after passing through **k** cross-/self-attention layers for the sentence and video clip, respectively. Depending on the task (see note below), we obtain these contextualized features $S_k$ and $C_k$ after passing the input feature matrices $S_0$, $C_0$ through one or more cross-/self-attention layers. Note that the feature representation for the **j**-th word in the sentence is denoted by column vector $S_k[:,j]$ (similarly $C_k[:,j]$ for the **j**-th spatiotemporal region in a video clip).

Following [17], we seek to contrast the contextualized sentence features $S_k$ against the input sentence features $S_0$.  To achieve this goal, we first pass the input features $S_0$ through a multilayer perceptron (MLP) to obtain projected “value” features $S^{val} = MLP(S_0)$. Given a negative set neg of contextualized word features that do not correspond to the video clip, we then contrast the projected value features $S^{val}$ against the contextualized features $S_k$ using the InfoNCE loss:
$Loss = \frac{\sum_{j=1}^N \exp(dot(S_k[:,j], S^{val}[:,j])}{\exp(dot(S_k[:,j], S^{val}[:,j]) + \sum_{S^{neg} \in neg} \exp(dot(S^{neg}, S^{val}[:,j]))}$

where dot(.) denotes the dot product operation and **N** denotes the number of features (columns) in matrices $S_0$ and $S_k$.

Note that our word loss is very similar to the one used in [17]. For fair comparison on the image grounding task, we apply the word loss to contextualized features that have passed through the first cross-attention layer only (as depicted in Fig 4(b)), which closely follows the setup in [17]. For the video task, we applied the word loss to the contextualized features after the final attention layer in the CoMMA module. We will update the paper to clarify this loss and are happy to incorporate further suggestions to improve clarity.

**Multiple CoMMA modules - Reviewers rFC4 and gTL6**: We have explored using 2 CoMMA modules in our experiments but due to the large memory requirements, the batch size had to be reduced significantly. The results from our initial experiments and prior work on contrastive learning show that a large batch size is crucial to achieving strong performance. We will clarify these details in the camera ready.

---

> ### Comment · Reviewer_rFC4 · 2021-08-24
> **Clarification on General Response**
>
> I am still confused by the Equation (6). I think the main question is: can you confirm that $S_k$ effectively contains features obtain from the visual modality and not the language (thanks to the cross attention module that is outputting features from the other modality). In particular I would believe that this property is crucial to avoid having a trivial contrastive loss. This should be clearly emphasized in the paper. I think notations participate to that confusion as $S$ is used both for input features that contain language features and also for features out of the first cross attention layer that now contains visual features (obtained via an attention mechanism where the queries come from language and keys and values come from vision).
>
> It might be worth coming up with a distinctive notation to clearly indicate that the features contain vision e.g. $S^v_k$ (not saying that is the best way to do, I am open to other suggestions).
>
> Please at least confirm the above question.

---

> > ### Author Response · Authors · 2021-08-25
> > **Authors' response**
> >
> > Thank you for your suggestions on the Equation 6 notation. We confirm that in our proposed CoMMA model, as well as in Equation 6, the outputs $S_K$ and $C_K$ contain contextual features from only one modality each. To clarify the confusion, we will modify the current notation to reflect the modality from which the output features are computed. Namely, we will use the suggested $S_K^V$ and $S_K^L$ notation to denote contextual features that are computed from the visual and language modalities, respectively. Here, $K$ can be used to represent a cross-attention or self-attention layer.
> >
> > Hence, the outputs of the intermediate layers in our proposed CoMMA model can be formalized as such:
> >
> > 1. $(S_{CA1}^V, C_{CA1}^L) = CA1(S_0^L, C_0^V)$
> > 2. $(S_{SA}^V, C_{SA}^L) = SA(S_{CA1}^V, C_{CA1}^L)$
> > 3. $(S_{CA2}^L, C_{CA2}^V) = CA2(S_{SA}^V, C_{SA}^L)$
> >
> > Where $(S_{CA1}^V, C_{CA1}^L)$ denotes the output of the first cross-attention layer, $S_{CA1}^V$ represents the contextualized features computed from the visual modality and $C_{CA1}^L$ represents the contextualized features computed from the text modality.

---

> > > ### Comment · Reviewer_rFC4 · 2021-08-26
> > > **Thanks**
> > >
> > > Thank you this is much clearer now.
> > >
> > > In my opinion, using this notation in the paper will highlight better that subtle and crucial point about the proposed approach.

---

### Decision · Program_Chairs · 2021-09-27

**Decision:**

Accept (Spotlight)

**Comment:**

This paper has a very nice insight, and its essence has been made much clearer by the discussion and new notation during the rebuttal discussions. The two reviewers who originally recommended rejection have upgraded their scores to accept following these discussions.

The authors have promised to make many updates to the paper, and should take note of the post-rebuttal statements of the reviewers.